# Exploring a Multi-Layered Cross-Genre Corpus of Document-Level Semantic Relations

Gregor Williamson *, Angela Cao †, Yingying Chen †, Yuxin Ji †, Liyan Xu and Jinho D. Choi *

Department of Computer Science, Emory University, Atlanta, GA 30322, USA
* Correspondence: gjwilliamson@hotmail.co.uk (G.W.); jinho.choi@emory.edu (J.D.C.)
† These authors contributed equally to this work.

**Abstract:** This paper introduces a multi-layered cross-genre corpus, annotated for coreference resolution, causal relations, and temporal relations, comprising a variety of genres, from news articles and children's stories to Reddit posts. Our results reveal distinctive genre-specific characteristics at each layer of annotation, highlighting unique challenges for both annotators and machine learning models. Children's stories feature linear temporal structures and clear causal relations. In contrast, news articles employ non-linear temporal sequences with minimal use of explicit causal or conditional language and few first-person pronouns. Lastly, Reddit posts are author-centered explanations of ongoing situations, with occasional meta-textual reference. Our annotation schemes are adapted from existing work to better suit a broader range of text types. We argue that our multi-layered cross-genre corpus not only reveals genre-specific semantic characteristics but also indicates a rich contextual interplay between the various layers of semantic information. Our MLCG corpus is shared under the open-source Apache 2.0 license.

**Keywords:** multi-layered corpora; cross-genre corpora; relation extraction

## 1. Introduction

In this paper, we present a multi-Layered cross-genre corpus of document-level semantic relations (MLCG). Our corpus is *multi-layered* as it is annotated for coreference resolution, causal relations, and temporal relations. It is *cross-genre*, consisting of text from news articles, children's stories, and Reddit posts about college life.

We conduct a quantitative analysis that reveals distinct characteristics for each genre across the three semantic layers. In particular, we observe that children's stories have a temporally linear narrative with a clear sequence of events. Stories are driven by third-party protagonists, with causal relations prominently featured. Stories contain more mentions than other data sources and a higher frequency of entity types due to repeated reference to the characters. In contrast, news reports have a mostly non-linear temporal structure. Although the majority of reported events are in the past, they are presented in order of importance rather than chronologically. Both first-person pronouns and conditional language are rare. Our Reddit posts, on the other hand, center around the author and their present situation. Causal language is used to describe and rationalize the author's circumstances. First-person pronouns are used with a high frequency, while generic pronouns and reference to the audience are also relatively common. Occasional reference is made to the situation as a whole, as well as meta-textual reference to the post itself. Together, these observations highlight the unique difficulties each genre presents for annotators and machine learning models.

The paper proceeds as follows. In Section 2, we provide background on both multi-layered and cross-genre corpora, focusing on the relations annotated in our corpus. In Section 3, we briefly present the annotation schemes adopted for each of these semantic relations. These schemes are based on existing schemes, which have been adapted

to suit the particular needs of the various text types. Section 4 provides a description of the corpus, annotators, tools, and the training employed. In Section 5, we present inter-annotator agreement (IAA) scores, as well as metrics that highlight the unique qualities of each genre. Section 6 highlights potential future directions. In Section 7, we release our corpus under the open-source Apache 2.0 license. Finally, Section 8 concludes.

## 2. Related Research

### 2.1. Multi-Layered Corpora

Multi-layered corpora are corpora that are annotated for multiple, mutually independent layers of natural language information on the same text [1–5]. While the contents of one layer may not be directly and immediately inferred from the contents of another, there may nonetheless be some correlation between elements of one layer and another. In this way, each layer may provide additional context for the others. Typically, multi-layered corpora are annotated for local, word-level, or sentence-level syntactic or semantic features such as parts of speech, named entity labels, or dependency structure, with the notable exception of coreference resolution, which is a staple of NLP research [6–8]. However, there is some recent work toward annotating multiple document-level semantic relations in a single corpus. For instance, the authors of [9] annotated a corpus of news stories for temporal and coreference relations, in addition to sentence-level semantic role labeling.

### 2.2. Cross-Genre Corpora

Traditionally, the majority of corpora have been comprised of newswire. However, this may lead to bias and an "overestimation of expected accuracy in both manual and automatic annotation" [5]. For this reason, there has been a concerted effort in recent years to move toward creating diverse corpora that contain texts from various genres or domains. Annotation schemes designed for such corpora, as well as ML models trained on such corpora, are likely to be more robust than those built around a single data source, as they are more capable of extending to novel domains. Existing work in this direction includes GUM [3], a multi-layer corpus that annotates various data sources using the same guidelines, exploring the effect of text types across news, interviews, travel guides, and how-to guides. We add to this stream of research by comparing the text type effects across news, children's stories, and Reddit posts from college subreddits. Reddit data in particular are less commonly annotated for document-level semantic information such as temporal and causal relations due to their noisy and uncurated nature. Applying the same annotation scheme to different data may also provide interesting insights into the textual characteristics of the data source. It is this final point that will be the central focus of our findings. Specifically, we demonstrate that the properties of each semantic layer differ from genre to genre, while also coming together to reflect the underlying characteristics of each text type.

### 2.3. Coreference Annotation

Coreference is a prevalent yet complicated phenomenon in natural language that requires an understanding of pragmatics, as it is highly dependent on context. An entity may be denoted by different forms, and the same form may be used to denote different entities in different contexts. Furthermore, coreference is unbounded, meaning document-level or discourse-level annotation is required. Consequently, coreference resolution is a long-standing task in NLP.

Despite numerous efforts made in the past few decades, current coreference resolution models still frequently encounter problems, partly due to the lack of well-annotated corpora. Specifically, existing gold standard corpora largely focus on well-edited texts. The current benchmark, OntoNotes [8], and prior efforts, including MUC [6] and ACE [7], contain a variety of news and broadcast data in multiple languages. After the launch of OntoNotes, there were various attempts at coreference resolution in different genres, including English literature [10], school examinations [11], Quiz Bowl questions [12], and biomedical reports [13].

## 2.4. Temporal Annotation

Semantic representation of temporal relations is an important task in the field of computational linguistics. Determining the number of events and the order in which these events happened is key to machine reading comprehension. Thus, the development of robust temporal dependency rules is central for constructing accurate timelines of events in a text.

The TimeML annotation scheme [14] lays out one of the earliest schemes for temporal relation annotation by anchoring event predicates, resulting in a corpus of annotated news articles: TimeBank. Evaluation of TimeML's efficacy was initially undertaken in TempEval-1 [15]. TempEval-2 expanded this earlier effort with three additional subtasks for identifying events and time reference across data in six languages [16]. Finally, TempEval-3 employed the most extensive dataset under the TimeML scheme, introducing a new metric to rank systems per subtask [17].

Despite these advancements, the TimeML-based approaches sometimes suffered from vague and context-sensitive definitions of relations and events. This prompted recent research to streamline these temporal annotation rules by focusing on non-overlapping elements. For instance, the authors of [18] introduced a multi-axis annotation scheme to focus on annotating event pairs that are considered relevant, while [19] proposed a dependency tree structure that allows every event to have a single reference time and more than one child event. The authors of [20] proposed a temporal dependency graph that better captures the completeness of temporal orderings compared to hierarchical dependency tree structures by allowing for multiple time references for a single event. The authors of [21] incorporated temporal relations as parts of Uniform Meaning Representations, in which before, after, contained, and overlap relations are used.

Temporal relation annotation has been conducted on children's stories [22], everyday life stories [23], news [9,14,18,24], and Wikipedia articles [25]. In this paper, we adapt the annotation scheme of [22] for large-scale annotation in disparate domains, outlining the challenges faced and providing characterizations of the semantic properties of each genre of text.

## 2.5. Casual Annotation

Often building on work in the temporal domain, further efforts have been made to create corpora of causal relations. Causal language has long been of interest to linguists, cognitive scientists, and computational linguists. Cognitive approaches based on the theory of force dynamics of [26], such as [27], argue that periphrastic causal verbs can be aspectually grouped into the types cause, enable, and prevent, or a combination of these. Table 1 summarizes the force dynamic theory of [27].

**Table 1.** Defining cause, enable, and prevent according to [27].

|         | Patient Tendency toward Result | Affector–Patient Concordance | Occurrence of Result |
|---------|:-----------------------------:|:----------------------------:|:--------------------:|
| cause   | N | N | Y |
| enable  | Y | Y | Y |
| prevent | Y | N | N |

Annotation projects such as [28,29] have adapted this categorization when developing causally annotated corpora. Here, the notion of causation supersets the type cause, which is in turn distinct from the lexeme *'cause'*. Other work, such as [30,31], define these concepts using computationally implementable formalisms. The authors of [32] provide a semantics for particular verbs such as *'affect'*, *'enable'*, and *'made no difference'* using a causal judgement task. Similar cognitive studies, such as [33,34], are especially interested in how people make judgements about causation. They propose that faulting a causal event takes into account both whether the cause affects *how* the effect occurs, as well as *whether* it did.

Consider the following examples of corpora that focus only on causal discourse relations. In ref. [35], causal discourse relations are annotated by predicate and argument, where ARGM-CAU is used to annotate "the reason for an action", as in '*they* [PREDICATE *moved*] *to London* [ARGM-CAU *because of the baby*]' [28]. More recently, the semantic annotation framework Causal and Temporal Relation Scheme (CaTeRS) was developed by [29]. This scheme is applied to 320 five-sentence short stories sampled from [23]'s ROCStories corpus. The CaTeRS framework annotated causal and temporal relations simultaneously, while distinguishing between the cause, enable, and prevent causal concepts.

The BECauSE corpus of causal relations implements the annotation schema developed in [36]. It includes annotated relations of 59 articles from the Washington section of the New York Times (NYT) corpus, 47 Wall Street Journal (WSJ) documents from the Penn Treebank, 12 documents from the Manually Annotated Sub-Corpus, and 772 sentences transcribed from Congress's Dodd–Frank hearings. The causal relations in this combined corpus were annotated based on pre-identified connectives, which directed ARGCs (causes) to ARGEs (effects). Notably, the author of [36] expresses a desire to attempt more fine-grained distinctions of cause, enable, prevent based on those in [27], as well as extending their annotation scheme to other relation types such as concession and comparison.

Most recently, the authors of [37] developed a cognitive approach to annotating cause, enable, prevent relations based on the work of [36,38]. We expand on this work here, adapting the approach to a broader range of texts.

## 3. Annotation Schemes

In this section, we briefly describe the schemes for annotating each of the semantic layers of our corpus. For comprehensive details of the annotation schemes, please refer to the guidelines available at https://github.com/emorynlp/MLCG (accessed on 27 July 2023).

### 3.1. Coreference

We followed the OntoNotes guidelines for coreference annotation in identifying mentions and establishing coreferential relationships. However, we made a number of adjustments to accommodate unique uses of pronouns found in the Reddit data. **Singletons:** Following the OntoNotes guidelines, we do not mark singletons, except for on two occasions. We discuss these two cases, `doc-situation` and `post`, below.

**Entity:** Noun phrases and pronouns denoting entity concepts are labeled as `entity`.

**Event:** We also annotate event concepts. However, event mention identification is a more challenging task than entity identification, as it covers diverse syntactic categories ranging from verbs to gerunds and noun phrases [39]. In hopes of differentiating entities and events conceptually, we label a mention with `event` as long as it refers to an event concept.

**Generic mentions:** We make a distinction between generic and specific mentions and follow PreCo's guidelines where generic mentions can directly corefer with one another. While OntoNotes does not annotate coreference between generic mentions, we find such relations quite common in Reddit posts, most notably with the generic third-person *'you'*.

**Doc-situation and post:** We annotate two mention types prevalent in Reddit data. Posts on subreddits about college tend to center around the author expressing their feelings and asking for help. The posts are relatively short posts describing ongoing situations (such as a problem), centered around the author, with frequent reference to the situation as a whole. Such reference is vague, and pronouns referring to document-level situations cannot be linked back to a single event span or even a set of spans. For example, in Figure 1, the first *'this'* refers to the whole situation causing the user stress instead of a specific event. To capture this, we add a new mention type called `doc-situation` for reference to vague document-level situations. These labels are often singletons. However, multiple occurrences of the same type can be linked by identity coreference just like standard cases of entity/event coreference. Another challenge specific to forum discussions is frequent reference to the post itself. While it is clear that there is self-reference to the document,

there is no span in the text for annotators to establish a coreference link, as shown in Figure 1. With this in mind, we introduced a standalone mention type called post for mentions referring to the document itself.

[...] I've now been left to fully reanalyze half of the experimental results presented and then rewrite a quarter of the thesis, which they somehow expect me to re-submit within 8 weeks. This ₍doc-situation₎ has to be every PhD's worst nightmare. I know I'm just too stressed. Thank you all so much for reading this ₍post₎.

**Figure 1.** Use of document situation and post mentions.

**Quantifier phrases:** Following the same rationale as [10], we annotate all quantifiers including negated existentials for consistency under situations like '[*No boy*] *took a picture of* [*himself*]'.

**Appositive, attributive, and expletive uses:** Unlike OntoNotes, which annotates appositive and attributive uses, we only focus on identity coreference for our corpus.

**Spans:** Unlike OntoNotes, which marks the maximum span, we only annotate the syntactic heads of noun phrases. This decision was made to keep span annotation consistent with the temporal layer.

**Subset:** In addition to identity coreference, we mark subset–set relations of entities and events like '[*the boy*] *got the lowest grade among* [*the students*]' in a separate layer. Subset relations involving generics are not annotated. The subset–set relation will link the two identity coreference chains if they exist, while at the same time allowing singletons to be involved.

*3.2. Temporal*

We based our temporal relation guidelines on the annotation scheme of [22]. Annotation of temporal relations is a three-part process. First, annotators identify the spans corresponding to events in the text. Next, they link each event to a reference time in the form of another event or the document creation time (DCT). Finally, annotators choose an appropriate temporal relation between these pairs.

**Document Creation Time (DCT):** In our corpus, the DCT is typically a dummy element at the beginning of each text that acts as the center of deixis for temporal reference. For the Reddit posts, this is the time that the author makes their initial post. Typically, posts have only one DCT. However, in some cases, the authors further edit the original Reddit post to give one or more updates. In these cases, the temporal reference in the update is anchored to a new time, which is after the original DCT. Thus, we instruct annotators to create multiple DCTs, ordered chronologically, when such a situation occurs.

**Event Identification:** Following [22], we only annotate events that contribute to the temporal narrative of the text. That is, we do not annotate events that occur in non-veridical environments such as speech or modal and hypothetical clauses, as well as under negation. We also extend this principle to questions, imperatives, exclamations, and exaggerations. In addition, we adopt the best-phrasing rule when annotating phrasal verbs. For instance, aspectual verbs such as '*start*', '*continue*', '*stop*', '*remain*', and '*let*' are not annotated. Instead, the verbal complements to these aspectual verbs are annotated, since it is this event that drives the temporal narrative of the text.

**Pair Identification:** Annotators are instructed to link the events together with directed edges. Typically, parent events proceed their dependents with the notable exception of sentence-initial dependent clauses. In this case, annotators are instructed to link the events in the main clause to the larger temporal structure, and then annotate the events in the dependent clauses as branching off the main clause event.

**Relation Identification:** Finally, annotators are instructed to identify the label of the relation that holds between a pair of events. The temporal relations employed here, shown in Table 2, are based on those in Uniform Meaning Representations [21].

**Table 2.** Table of temporal relations.

| Relation | Definition |
| --- | --- |
| *A* before *B* | event *A* finished before event *B* started |
| *A* after *B* | event *A* started after event *B* finished |
| *A* contains *B* | the run time of event *A* contains the time of event *B* |
| *A* contained-in *B* | the run time of event *A* is contained in the time of event *B* |
| *A* overlap *B* | the run times of events *A* and *B* overlap |

*3.3. Causal*

Our causal annotation scheme is based on that of [36–38]. This scheme focuses on causal categories such as Purpose, Motivation, and Consequence. We aim to extend the applicability of these tools to categorize cause, enable, and prevent types of causation. We adapt a modified version of the *Constructicon* from [36], sample entries of which are shown in Figure 2.

| Connective pattern (verbs given in present tense and nouns/adjectives given as copulas for readability) | Variants (not including passives, infinitives, or nominalizations of verbs) | Words to annotate as connective ([ ] = may not be present) |
| --- | --- | --- |
| <cause> forbids <effect> | <cause> forbids <effect> to <effect> | forbid, [to] |
| <cause> prevents <effect> | <cause> prevents <effect> from <effect> | prevent, [from] |
| <cause> prohibits <effect> | <cause> prohibits <effect> from <effect> | prohibit, [from] |

**Figure 2.** Example causal constructions in the Constructicon.

The Constructicon is a resource for annotators that stores around 200 pre-identified causal connectives such as *'because'*, *'allow'*, and *'after'*. Annotators are tasked with searching for these connectives in a text in order to identify an instance of causal language. Following [37], we provide annotators with the Causal Relations Decision Tree (CRDT) to reduce the cognitive load and subjective variability in determining causal category. The decision tree is a flowchart designed to ground the notions of cause, enable, and prevent so that annotators are not burdened with the task of internalizing these vague and abstract concepts. This flowchart can also be found in our guidelines https://github.com/emorynlp/MLCG, (accessed on 27 July 2023). While these tests are not definitive, they aid in systematizing intuitions that previous researchers have expressed about the concepts [27]. For more details on the annotation scheme adopted here, see [37].

**4. Materials and Methods**

*4.1. Data*

Our dataset is drawn from three distinct sources: (i) Children's stories consisting of fables ($n = 50$) and excerpts from the Wizard of Oz (Project Gutenberg https://www.gutenberg.org/, (accessed on 15 February 2022)) ($n = 50$); (ii) news stories from CNN (The cnn_dailymail corpus https://huggingface.co/datasets/cnn_dailymail, (accessed on 15 February 2022)) ($n = 50$) and Reuters (NLTK dataset https://www.kaggle.com/datasets/nltkdata/reuters, (accessed on 15 February 2022)) ($n = 50$); and (iii) Reddit posts from subreddits about college (Reddit college dataset https://github.com/emorynlp/reddit-college, (accessed on 15 February 2022)) ($n = 100$). An additional 50 excerpts from Wind in the Willows (Project Gutenberg (accessed on 15 February 2022)), and 50 Reddit posts are annotated for temporal relations. Lastly, an additional 10 Reddit posts that were used during training are also included. A break down of the data sources is provided in Table 3.

**Table 3.** Corpus composition by data source.

|  | Causal | Coref | Temporal |
|---|---|---|---|
| CNN | 50 | 50 | 50 |
| Fables | 50 | 50 | 50 |
| Reddit | 110 | 110 | 160 |
| Reuters | 50 | 50 | 50 |
| Wind in the Willows | - | - | 50 |
| Wizard of Oz | 50 | 50 | 50 |
| **Total** | **310** | **310** | **410** |

All data are tokenized using the ELIT Tokenizer (https://github.com/emorynlp/elit-tokenizer, (accessed on 16 February 2022)) and filtered to a length of between 100 and 200 tokens ($100 < n < 200$). This range is used to allow for a sufficient number of fables, which are relatively short in length. The Reddit posts consist of the post itself and any edits; comments and replies are not included. The Reddit posts are also filtered to remove posts containing profanity using the Profanity-Check Python library (https://github.com/vzhou842/profanity-check, (accessed on 16 February 2022)). Table 4 gives a breakdown of the total number of tokens and sentences for each genre in the full corpus, the average document length in tokens and sentences, and the average sentence length in tokens. Reddit documents have the shortest sentences, and therefore have the most sentences per document, while stories have the longest sentences, and thus have fewest sentences per document.

**Table 4.** Token and sentence count, average document length in tokens and sentences, and average sentence length in tokens for each genre.

|  | News | Reddit | Stories | Total |
|---|---|---|---|---|
| Documents | 100 | 160 | 150 | 410 |
| Tokens | 14,442 | 22,668 | 21,412 | 58,522 |
| Sentences | 606 | 1140 | 851 | 2597 |
| Tokens per document (avg.) | 144.42 | 141.68 | 142.75 | 142.74 |
| Sentences per document (avg.) | 6.06 | 7.13 | 5.67 | 6.33 |
| Tokens per sentence (avg.) | 23.83 | 19.88 | 25.16 | 22.53 |

A word cloud and top-10 unigram count for each genre is provided in Appendix A Figures A1–A3.

### 4.2. Training

To ensure that annotators meet a certain standard, they undergo extensive annotation training. For each scheme they annotate, they are required to (i) read the guidelines https://github.com/emorynlp/MLCG, (accessed on 27 July 2023), (ii) watch an instructional video, (iii) take one or more online tests, and (iv) annotate 10 test documents.

### 4.3. Annotation

Annotation was performed using the INCEpTION annotation tool (https://inception-project.github.io/, (accessed on 21 February 2022)). Figure 3 shows an example of causal annotation in the INCEpTION user interface.

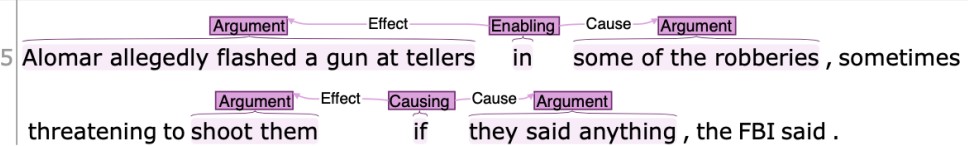

**Figure 3.** Example of causal annotation using the INCEpTION tool. Image shows annotation of the fifth sentence in a news document. Span labels represent cause type, edge labels represent argument type.

Although all annotators undergo training, they are also instructed to rotate through the various data sources in batches of 5 to ensure that any difference in IAA scores is not a result of familiarity with the annotation tool or experience following the annotation scheme.

Annotators include 7 paid undergraduate students as well as 4 authors of the guidelines (total $n = 11$). If annotators perform poorly during training for one of the schemes, they are excluded from annotating, leaving only 3 annotators for causal relations, 4 for temporal annotation, and 5 for coreference (one of the authors annotated both coreference and temporal relations).

## 5. Results

In this section, we analyze the various data sources. We observe a number of asymmetries between the different genres of text. The findings for each layer of annotation converge to reflect the underlying characteristics of each text type. For instance, children's stories are temporally linear, driven by third-party protagonists, with frequent causal relations. News data rarely feature first-person pronouns or conditional language, and the temporal ordering is often non-linear; events are typically reported in order of importance rather than chronology. On the other hand, Reddit posts are normally about the author and their current situation. The causal language centers around explanations for the current situation and intentions for how to change or improve it. There is frequent use of generic pronouns, frequent reference to the audience, occasional reference to the situation as a whole, and even meta-textual reference to the post itself.

To test the robustness of the annotation guidelines, we doubly annotate a portion of the dataset in order to evaluate inter-annotator agreement (IAA). For each layer, we start by reporting the annotator agreement scores before moving onto a quantitative analysis of each genre's characteristics.

### 5.1. Coreference Statistics

#### 5.1.1. Inter-Annotator Agreement

Table 5 shows IAA scores for coreference relations across the three genres. The quality of annotation is measured through the standard evaluation metrics for coreference, including MUC [40], $B^3$ [41], and CEAF$_{\phi4}$ [42]. The IAA for coreference across all metrics is highest for children's stories, with scores of 88.75 (MUC), 83.12 ($B^3$), and 75.49 (CEAF$_{\phi4}$), with an average of 82.45. The Reddit posts also perform well, with scores of 85.29 (MUC), 80.21 ($B^3$), and 71.61 (CEAF$_{\phi4}$), resulting in an average of 79.04. The IAA for news has a relatively lower score compared to the other genres, with scores of 68.73 (MUC), 65.43 ($B^3$), and 64.70 (CEAF$_{\phi4}$), with an average of 66.29. Overall, the results show good performance in coreference annotation for all three document types, with children's stories displaying the highest performance.

**Table 5.** Comparison of coreference performance on different text types annotated using the same guidelines.

| Text Type | MUC | $B^3$ | CEAF$_{\phi4}$ | Avg. |
|-----------|-------|-------|----------------|-------|
| News | 68.73 | 65.43 | 64.70 | 66.29 |
| Reddit | 85.29 | 80.21 | 71.61 | 79.04 |
| Stories | 88.75 | 83.12 | 75.49 | 82.45 |

#### 5.1.2. Analysis

Table 6 shows the count of the mention types across the three text types. While all three types of data contain mostly entities, children's stories have the highest entity frequency and more mentions per document. This is explained by the text type's story-telling purpose with frequent reference to the characters. This characterization is supported by the unigram analysis and word cloud representation in Figure A3, where 3 of the top 10 unigrams are character names ('*Dorothy*', '*Mole*', and '*Rat*'). News articles have more event labels than the other text types, which aligns with our intuition that news tends to refer to events that

happened in the past. These events often occur in coreference chains, since news reports often reiterate the story highlighting different information with each retelling. Reddit data contain a much more varied selection of mention types than the other two text types. In particular, Reddit contains more generic terms such as the generic *'you'* and *'people'* in general to describe certain situations.

**Table 6.** Count of mention types in different text types.

| Category | n | News Freq. | Mean | n | Reddit Freq. | Mean | n | Stories Freq. | Mean |
|---|---|---|---|---|---|---|---|---|---|
| entity | 1497 | 88.6% | 14.97 | 1947 | 86.8% | 17.70 | 2241 | 96.2% | 22.41 |
| event | 143 | 8.5% | 1.43 | 167 | 7.4% | 1.52 | 56 | 2.4% | 0.56 |
| generic | 42 | 2.5% | 0.42 | 95 | 4.2% | 0.86 | 32 | 1.4% | 0.32 |
| doc-situation | 0 | 0.0% | 0.00 | 27 | 1.2% | 0.25 | 0 | 0.0% | 0.00 |
| post | 7 | 0.4% | 0.07 | 6 | 0.2% | 0.05 | 1 | 0.04% | 0.01 |
| Total | 1689 | 100% | 16.89 | 2242 | 100% | 20.38 | 2330 | 100% | 23.30 |

Among the two labels deliberately introduced to accommodate the Reddit data, `doc-situation` is only presented in Reddit texts. While the news reports and children's stories might also describe a situation, they do not have much demand to refer back to that situation. Surprisingly, the `post` label, which was introduced specifically for the Reddit data, is also present in the news data. In the news texts, the mentions labeled as `post` are of the form *'this report'*, referring to the news document itself. In Reddit, the form of the `post` mentions are predominantly pronouns such as *'this'* or *'it'*, and occasionally noun phrases like *'this post'*.

Table 7 shows the count and frequency of first-person pronouns across text types. Over half of the pronouns (69.4%) involved in a coreference chain in the Reddit data are first-person pronouns, while only nine first-person pronouns (2.9%) are involved in coreference in the news data.

**Table 7.** Count of pronouns in different text types.

| Category | n | News Freq. | Mean | n | Reddit Freq. | Mean | n | Stories Freq. | Mean |
|---|---|---|---|---|---|---|---|---|---|
| 1st pers. | 9 | 2.9% | 0.09 | 1081 | 69.4% | 9.83 | 204 | 15.2% | 2.04 |
| All pronouns | 311 | 100% | 3.11 | 1557 | 100% | 14.15 | 1346 | 100% | 13.46 |

Table 8 shows the average number of coreference chains and their length. While news has the highest number of chains per document (6.41), the average coreference chain length is only between two and three mentions. News also has the largest distance of a mention to its nearest antecedent of nearly 30 tokens, around double that of Reddit and the children's stories. This reflects the fact that news is often reported in a disjoint fashion. Events are described in order of importance, with frequent repetition used as a means of adding further information. The children's stories have the longest chains, which reach an average of around four mentions.

**Table 8.** Statistics on coreference annotation in different text types. Column '1st PP' shows statistics for first-person pronouns.

| | News Overall | 1st PP | Reddit Overall | 1st PP | Stories Overall | 1st PP |
|---|---|---|---|---|---|---|
| Chain per document (avg.) | 6.41 | 0.02 | 5.50 | 1.09 | 5.93 | 0.68 |
| Mention per chain (avg.) | 2.64 | 4.5 | 3.72 | 9.01 | 3.94 | 3.00 |
| Characters per mention (avg.) | 10.45 | 1.33 | 4.30 | 1.35 | 5.90 | 1.50 |
| Token distance to antecedent (avg.) | 29.58 | 10.25 | 16.04 | 10.85 | 14.87 | 9.20 |

Given the strikingly large proportion of first-person singular pronouns in Reddit data, we explore the coreference chain data for the first-person singular pronouns as well. Identity chains where more than 75% of the mentions are first-person singular pronouns are considered first-person pronoun chains. In Reddit, coreference chains of first-person singular pronouns are over two times longer than the average chain length and occur in nearly every document. This discovery coincides with the intuition that Reddit data are closer to language used during everyday communication, which is often conducted from a first-person perspective. Such an intuition is supported by the unigram counts in Figure A2, where the verbs '*feel*', '*know*', and '*want*' are strongly represented, highlighting the personal, author-centered narrative of the posts.

5.1.3. Challenges

The language used in Reddit posts is mostly informal and colloquial. Most of the existing gold-standard coreference data focus on well-edited text types such as newspaper articles and novels, which typically consist of professionally written and carefully edited text. Internet abbreviations such as '*idk*' for "*I don't know*" make it impossible to mark the first-person pronoun inside the abbreviation alone using token-based mention identification. Similarly, missing punctuation, such as writing "*I'll*" as '*Ill*', can interfere with appropriate span identification.

Besides grammatical errors and abbreviations, another problem of Reddit data is the absence or unclear use of quotation marks. Direct quotation often involves a change in perspective and needs to be dealt with carefully for coreference. The misuse of quotation marks can cause problems for annotators trying to correctly identify coreference. While it is often possible for annotators to resolve coreference based on the context, the absence of a clear cue like that provided by quotation marks can make the challenge tougher for machine learning.

*5.2. Temporal Statistics*

5.2.1. Inter-Annotator Agreement

Table 9 shows IAA for event identification and temporal relation identification across the three genres measured using Krippendorff's $\alpha$. All semantic triples are normalized prior to evaluation (e.g., $B$ after $A \Rightarrow A$ before $B$).

**Table 9.** Inter-annotator agreement (Krippendorff's-$\alpha$) across genres.

|  | News | Reddit | Stories |
| --- | --- | --- | --- |
| Events | 0.86 | 0.75 | 0.85 |
| Relations | 0.56 | 0.47 | 0.48 |

It is clear that Reddit annotation poses the most significant challenge for annotators. Indeed, even the event identification task appears to prove difficult for annotators. A plausible explanation for this could be that Reddit posts frequently lack any indication of direct speech, as discussed above. The news data also show a notably higher agreement score for relation identification. The news reports often describe a simple sequence of events, which is retold with additional information added on each iteration. This is corroborated by the coreference findings, for which we observed a higher frequency of event coreference as the sequence of events is retold.

5.2.2. Analysis

Table 10 shows the average number of temporal points described per document across the three genres. Both the story and news data contain only a single deictic temporal anchor, annotated as DCT, per document. The Reddit texts, on the other hand, occasionally include edits or updates in the post. These edits occur after some time, and are typically framed against a later temporal anchor. As a result, the Reddit data have more than one DCT per document on average. The children's stories describe a linear sequence of

events. Consequently, they feature the greatest number of events per document. News reports, on the other hand, often contain reports of things that people have said. This is evidenced in the unigram count and word cloud representation in Figure A1, where the most represented token in the dataset is the word '*said*'. As such, there are notably fewer events per document.

**Table 10.** Average reference time count per document.

|  | DCTs per Document (avg.) | Events per Document (avg.) |
|---|---|---|
| News | 1.00 | 7.57 |
| Reddit | 1.16 | 10.32 |
| Stories | 1.00 | 11.54 |

Table 11 reports the percentage of relations by type (after normalization) across the three genres. It also shows what percentage of each relation type is anchored to the DCT. Reddit posts are typically about the author's ongoing real-world situation. They describe states or events that are ongoing at time the post is written. Predictably, they have the highest percentage of contains/contained-in relations (37.55%), nearly all of which are anchored to the DCT (90.97%). Compare this to the news data, which contain many deictic past tenses, evidenced by the high percentage of before/after relations (68.51%). Most of these are connected to the DCT (79.31%). This captures the intuition that events are often restated or told in a piece-wise fashion, rather than being connected within a linear narrative. Children's stories, on the other hand, are very linear in nature. They also have a relatively high percentage of before/after relations (45.11%). However, in stories, events are typically presented following one another. A deictic past time is established, and then subsequent events are linked to previous events. For this reason, we see the lowest percent of relations connected to the DCT. We also see the highest number of overlap relations in the stories. This is likely due to adverbial and temporal clauses being more prevalent than in the other genres.

**Table 11.** Representation of relation type across genre, and % of relation type connected to the DCT.

|  | News | | Reddit | | Stories | |
|---|---|---|---|---|---|---|
|  | % of Rels | % DCT | % of Rels | % DCT | % of Rels | % DCT |
| before/after | 68.51% | 79.30% | 31.68% | 58.47% | 45.11% | 19.80% |
| contains/contained-in | 13.46% | 82.14% | 37.55% | 90.97% | 4.98% | 7.21% |
| overlap | 18.03% | 0.00% | 30.77% | 0.00% | 49.91% | 0.00% |

## 5.3. Causal Results

### 5.3.1. Inter-Annotator Agreement

As demonstrated in Table 12, our overall corpus of causal annotations yields an $F_1$ score of 0.77 for connective identification excluding cases of partial overlap, which is improved from the 0.70 of [36]. When calculating our $F_1$ scores, we concatenated our documents into a single text before scoring. This was due to the irregular appearances of connectives; while some documents contained upwards of a dozen instances of causal connectives, there were also 22 of our 300 doubly annotated documents that did not feature any annotations at all. Furthermore, for agreed connective spans, the corpus also yielded a $\kappa$ score of 0.83 for types of causation. This is comparable to the 0.80 of [36] for the causation categories of Purpose, Motivation, and Consequence. However, our overall span score was lower than [36], at 0.75. This was likely due to argument length disagreement, as all three document types contained arguments with numerous modifiers.

**Table 12.** Comparison of causal relation annotation performance on different text types using the same guidelines. $\kappa$ indicates Cohen's kappa, which was only calculated for agreed spans (in line with [36]).

|  | News | Reddit | Stories | Overall |
|---|---|---|---|---|
| Spans ($F_1$) | 0.74 | 0.81 | 0.72 | 0.75 |
| Argument labels ($\kappa$) | 0.86 | 0.93 | 0.91 | 0.90 |
| Connective spans ($F_1$) | 0.75 | 0.82 | 0.75 | 0.77 |
| Types of causation ($\kappa$) | 0.89 | 0.78 | 0.82 | 0.83 |

5.3.2. Analysis

The analysis of our annotated corpus provides some interesting insights. Firstly, Table 13 and Figure 4 show the coverage of each type of causal type across the genres in our dataset. The most notable observation to be made is that cause-type instances dominate the annotated causal language. This is to be expected; the Causal Relation Decision Tree in our guidelines, which tests for cause-type causation, asks annotators whether the textual context *presents* the cause as necessary and sufficient for the effect. In the limited context of a 200-token Reddit post, authors use causal language to identify and point out causal relationships, thus delimiting the cause as contextually necessary and sufficient in some way for the effect to occur. Interestingly, our news data show significantly fewer instances of causal language. Moreover, the cause-type connectives make up a smaller portion of the casual language present. This makes some sense, as the role of news is to report events as they happened, and not to ascribe potentially speculative causal relations between them. Indeed, news writers may be careful to avoid implying any causal relation at all between events, instead leaving it up to the reader to infer such a relation.

**Table 13.** Counts of cause type across different text types.

|  | News | | Reddit | | Stories | | |
|---|---|---|---|---|---|---|---|
|  | *n* | Percent | *n* | Percent | *n* | Percent | Total |
| cause | 153 | 71.94% | 204 | 85.36% | 199 | 81.56% | 556 |
| enable | 37 | 24.90% | 28 | 11.72% | 36 | 14.75% | 101 |
| prevent | 9 | 3.16% | 7 | 2.93% | 9 | 3.69% | 25 |
| **Total** | 199 | 100% | 239 | 100% | 244 | 100% | 682 |

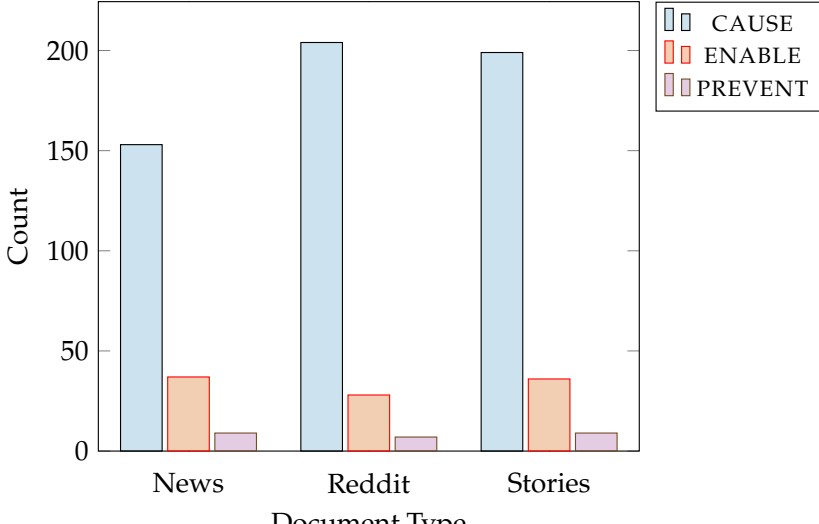

**Figure 4.** Bar graph of causal counts as presented in Table 13.

Next, consider Table 14, which depicts the most popular connectives across the different document types. Our findings align closely with those of [36]'s counts of connective patterns in the BECauSE corpus. However, it is interesting to note that their frequencies vary significantly across document types. For example, the conditional connective '*if*'

is only used causally five times in the news documents, reflecting the factual and non-hypothetical nature of reporting news. Consider also that '*after*' appears as our seventh most popular connective, despite only occurring in its causal use five times in the Reddit posts and children's stories combined. This is due to its frequent appearance as a causal connective in the news data. A similar but less pronounced situation holds for '*when*'. Compare this to the connectives '*because*' and '*so*'. Both occur frequently in the Reddit data, with '*so*' also appearing frequently in the stories. However, these connectives are rarely used in the news data. These data all support the observation that the news refrains from using obvious causal language when describing a sequence of events, opting instead to use temporal language as a means of implying causality.

**Table 14.** Comparison of popular connectives across different document types.

| Connective | News | | Reddit | | Stories | | Total | |
|---|---|---|---|---|---|---|---|---|
| | *n* | Freq. | *n* | Freq. | *n* | Freq. | *n* | Freq. |
| *for* | 29 | 14.57% | 28 | 11.72% | 45 | 18.44% | 102 | 14.96% |
| *to* | 29 | 14.57% | 32 | 13.39% | 34 | 13.93% | 95 | 13.93% |
| *if* | 5 | 2.51% | 23 | 9.62% | 26 | 10.66% | 54 | 7.92% |
| *because* | 3 | 1.51% | 44 | 18.41% | 4 | 1.64% | 51 | 7.48% |
| *so* | 2 | 1.01% | 22 | 9.21% | 22 | 9.02% | 46 | 6.74% |
| *when* | 13 | 6.53% | 11 | 4.60% | 20 | 8.20% | 44 | 6.45% |
| *after* | 26 | 13.07% | 4 | 1.67% | 1 | 0.41% | 31 | 4.55% |
| **Total** | 109 | 54.77% | 164 | 68.62% | 152 | 62.30% | 423 | 62.02% |

The fact that causal relations are rarely conveyed using the temporal connective '*after*' and '*when*' in the Reddit posts on college subreddits is in accordance with our observations made in the temporal layer. Namely, these Reddit posts describe ongoing situations at the time of writing, as opposed to a temporally ordered sequence of events. Finally, it is worth noting that the top seven connectives account for almost two-thirds of all causal uses of connectives in our corpus. This makes it even more surprising that certain connectives occur so frequently in some text types while being almost entirely absent in others.

*5.4. Summary*

The above subsections have described our findings layer by layer. In this section, we briefly summarize the landscape of each genre by describing their document-level characteristics. Firstly, we observe that children's stories are temporally linear narratives with a clear sequence of events. They are driven by third-party protagonists. Causal relations are prominent, reflecting clear cause-and-effect relationships. Furthermore, they have a high frequency of entities and more mentions per document due to frequent references to the story's characters.

News reports, on the other hand, feature largely non-linear temporal ordering. Events are reported in order of importance rather than chronologically. They rarely feature first-person pronouns or conditional language. They have the greatest number of event labels per document, indicating a focus on reporting past events. Finally, they have a frequent use of before/after relations, with a significant proportion anchored to the document creation time (DCT), indicating deictic uses of the past rather than a linear narrative.

Lastly, Reddit posts taken from college subreddits tend to focus on the author's feelings and desires and their current situation. The causal language centers around explanations for the current situation and the authors intentions for improving it. There is frequent use of generic pronouns and references to the audience. There is occasional reference to the situation as a whole and meta-textual reference to the post itself. There is also a very high frequency of first-person pronouns, often occurring in coreference chains. These findings are represented in Figure 5 below.

Overall, our results highlight the distinctive characteristics of each genre in terms of temporal ordering, causal language, and use of pronouns. These findings support the notion that the characteristics of the text type significantly influence each semantic layer. This emphasizes the importance of considering genre-specific features in multi-layered semantic annotation, as well as highlighting the need for cross-genre corpora in order to have a dataset that is representative of different domains.

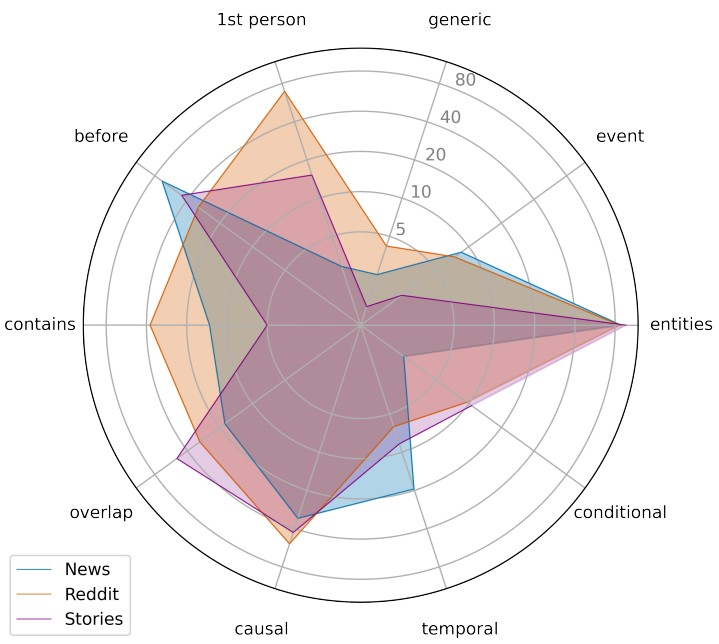

**Figure 5.** Visual representation of genre characteristics. Coreference: 1st person, generic, event, and entities. Causal: conditional connectives (*'if'*), causal connectives (*'for'*, *'to'*, *'because'*, *'so'*), and temporal connectives (*'when'*, *'after'*). Temporal: overlap, contains/contained, and before/after. All percentages are from the tables above. Graph is scaled logarithmically.

## 6. Discussion

Our findings highlight the need for a diverse range of data when annotating document-level semantic relations. This is especially relevant when training robust ML models. In future work, we plan to train a relation extraction model on the semantic relations annotated in our corpus. Each layer of annotation provides information to the others. In our dataset, certain textual properties across various layers are shown to be more prevalent across different genre types. We also hypothesize that models trained on diverse data sources such as these will be more robust in new domains. If a model is trained on news data alone, for instance, it will be primed to expect certain temporal, coreference, or causal relations that may be underrepresented or absent in domains such as Reddit posts or children's stories.

We see three directions for future research in this space. This type of quantitative characterization of semantic relations could be extended to other genres [2–4]. Alternatively, additional semantic or pragmatic relations could be annotated at both the sentence and document level [9]. This will help more appropriately characterize text types, and detect broader correlations between the various semantic layers. Finally, this approach can be extended to other languages [1,43]. This adds an additional dimension to a multi-layered cross-genre corpus. Efforts in this direction bring with it the complication of achieving annotation consistency across languages for each layer, requiring meaning representations that are universally applicable [21].

### 7. Corpus Release

We release MLCG in `.json` format under the Apache 2.0 license at https://github.com/emorynlp/MLCG, (accessed on 27 July 2023). In the temporal corpus, inverse relations are normalized. The temporal dataset is also provided with closure under entailment [44], which can lead to improved machine learning [45]. The closure rules are described below.

1.  *A* before *B* ∧ *B* before *C* ⇒ *A* before *C*
2.  *A* before *B* ∧ *B* contains *C* ⇒ *A* before *C*
3.  *A* contains *B* ∧ *B* overlaps *C* ⇒ *A* overlaps *C*
4.  *A* contains *B* ∧ *B* contains *C* ⇒ *A* contains *C*
5.  *A* contains *B* ∧ *C* contains *B* ⇒ *A* overlap *B*

### 8. Conclusions

In this paper, we have presented a multi-layered, cross-genre corpus of newswire, Reddit posts, and children's stories. Our corpus is annotated for three document-level semantic relations; coreference, causal relations, and temporal relations. Our annotation schemes are adapted from existing work to better apply to diverse data types. This saw the introduction of novel mention types to handle unique forms of reference in Reddit posts. Most significantly, however, we have shown that the semantic characteristics of different genres can vary significantly. In addition, we have seen that while the contents of our three semantic layers are mutually independent to an extent, they nonetheless correlate with one another to reveal broad textual characteristics that track our qualitative intuitions about each data source. This highlights the importance of not only annotating a diverse range of text types, but also annotating those text types for a broad range of natural language information. Without multi-layered annotation, rich contextual information remains hidden, and interesting interplay between the various levels of representation is lost. Likewise, without annotating semantic representations from a diverse selection of genres, annotation schemes may struggle when extended to new domains, and models trained on a poverty of genres will likely be biased. For these reason, we expect multi-layered, cross-genre corpora to grow ever more popular as their true value is recognized.

**Author Contributions:** Conceptualization, J.D.C. and G.W.; methodology, G.W., A.C., Y.C. and Y.J.; formal analysis, G.W., A.C., Y.C. and Y.J.; investigation, G.W.; resources, A.C., Y.C. and Y.J.; data curation, G.W.; writing—original draft preparation, G.W., A.C., Y.C. and Y.J.; software, L.X.; supervision, J.D.C.; project administration, J.D.C. and G.W.; funding acquisition, J.D.C. All authors have read and agreed to the published version of the manuscript.

**Funding:** This research was funded by the Amazon Alexa AI grant.

**Data Availability Statement:** Our annotated corpus and annotation guidelines are publicly available under the Apache 2.0 license on https://github.com/emorynlp/MLCG, (accessed on 27 July 2023).

**Acknowledgments:** We gratefully acknowledge the support of the Amazon Alexa AI grant. Any opinions, findings, and conclusions or recommendations expressed in this material are those of the authors and do not necessarily reflect the views of Alexa AI.

**Conflicts of Interest:** The authors declare no conflict of interest. The funding sponsors had no role in the design of the study; in the collection, analyses, or interpretation of data; in the writing of the manuscript, and in the decision to publish the results.

## Appendix A

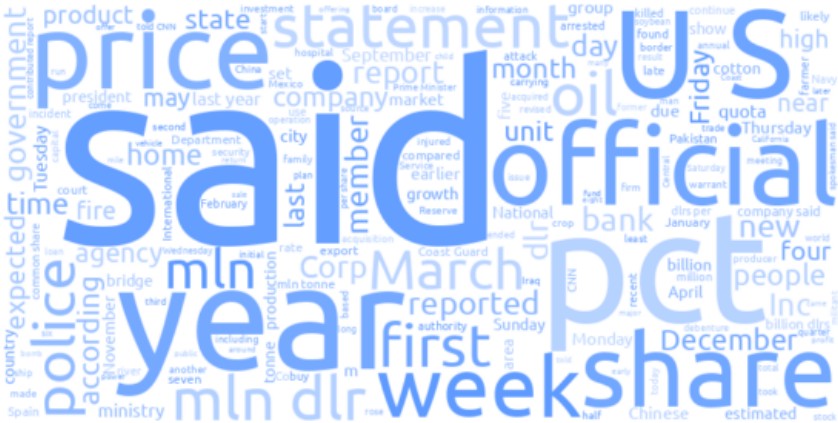

**Figure A1.** Word cloud for news. Unigram counts: '*said*' (249), '*year*' (74), '*pct*' (61), '*mln*' (60), '*dlrs*' (55), '*share*' (45), '*oil*' (36), '*U.S.*' (35), '*official*' (31), '*price*' (29).

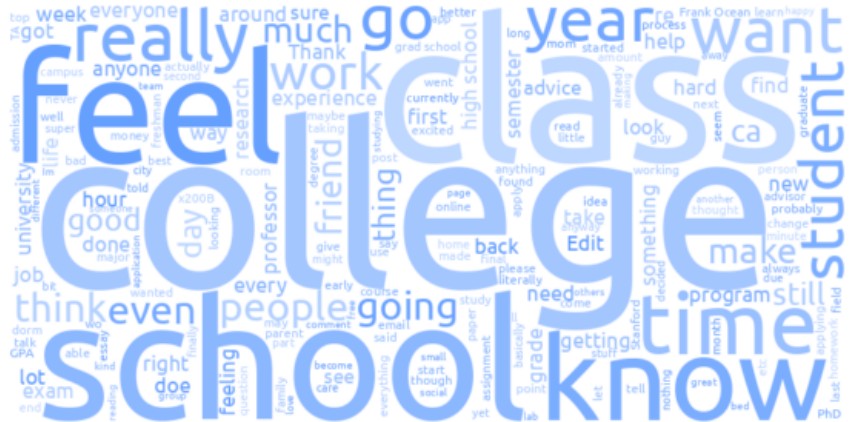

**Figure A2.** Word cloud for Reddit. Unigram counts: '*school*' (249), '*college*' (107), '*like*' (107), '*class*' (87), '*feel*' (76), '*year*' (72), '*know*' (68), '*time*' (66), '*really*' (63), '*want*' (59).

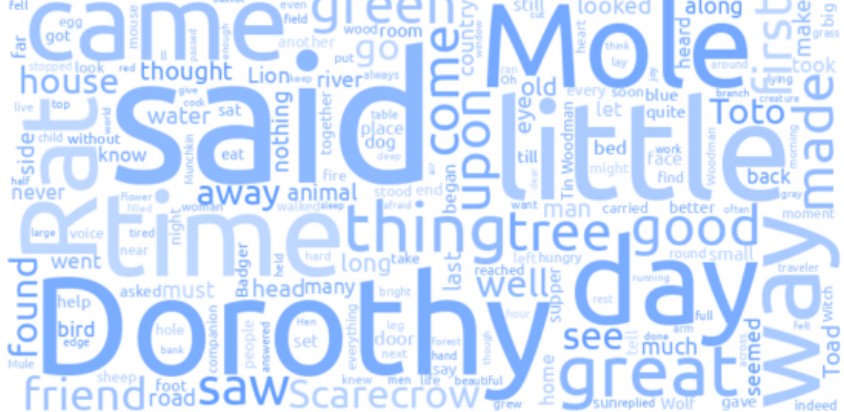

**Figure A3.** Word cloud for stories. Unigram counts: '*said*' (111), '*little*' (61), '*Dorothy*' (60), '*day*' (43), '*Mole*' (43), '*time*' (42), '*came*' (36), '*way*' (35), '*upon*' (33), '*Rat*' (33).

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
