# Peer review of "Exploring a Multi-Layered Cross-Genre Corpus of Document-Level Semantic Relations"

_information, doi:10.3390/info14080431_

Round 1

Reviewer 1 Report

The article focuses on the author's multi-layered cross-genre dataset. The dataset's annotation makes it applicable to various tasks, including coreference resolution, identifying causal relationships, and temporal dependencies. The authors not only provide information about genre-specific features but also discuss the challenges associated with manual annotation as well as annotation using machine learning models. It is noteworthy that the article effectively presents all the characteristics of multi-layered datasets. Furthermore, the authors take into consideration the language peculiarities of specific domains or genres when developing their dataset. Overall, the presented corpus and the accompanying article are highly valuable for Information readers.

There are a couple of points that I would like to clarify:

1. What are the prospects of creating a similar multilingual dataset? Are there possibilities for cross-lingual transfer (similar to MLQA, for example; see https://github.com/facebookresearch/mlqa)?

2. Most developers of modern datasets evaluate their datasets on task-specific baselines. Has such an evaluation been conducted for your dataset? This evaluation would be very useful for comparing the dataset with analogues for specific tasks. The presentation of results would be much more informative if the authors had evaluated the effectiveness of some baselines on the MLCG dataset and compared the obtained accuracies with researchers' results on other datasets.

3. It may be worthwhile to submit this article to the MDPI Data rather than Information.

The article is well-written, but there are a few places where articles, commas, and minor errors are missing. Please go through the text one more time

Author Response

Response:
1. What are the prospects of creating a similar multilingual dataset? Are there possibilities for cross-lingual transfer (similar to MLQA, for example; see https://github.com/facebookresearch/mlqa)?
- This is a great point for which we thank the reviewer. We have added a paragraph on this in the discussion.

2. Most developers of modern datasets evaluate their datasets on task-specific baselines. Has such an evaluation been conducted for your dataset? This evaluation would be very useful for comparing the dataset with analogues for specific tasks. The presentation of results would be much more informative if the authors had evaluated the effectiveness of some baselines on the MLCG dataset and compared the obtained accuracies with researchers' results on other datasets.
- This is similarly an excellent point. We have not yet conducted such an evaluation, but we hope to do so. Once that has been done we can upload it to the github repo's readme.

3. It may be worthwhile to submit this article to the MDPI Data rather than Information.
- We were invited to submit the manuscript to the Special Issue "Information Extraction and Language Discourse Processing" of MDPI Information by the editors.

Changes:
- Added a paragraph to discussion to address adding a cross-linguistic dimension to the corpus.
- Added bar graph Figure 4 to section 5.3.2.
- Added explanation that Reddit comments are not included with the posts (end of section 4.1).
- Information is added about multiple DCTs introduced during updates or edits (section 5.2.2). 
- An extra column is added in Table 9 to distinguish event times and DCT times.
- Table 4 is added. This shows the average number of tokens and number of sentences per document for each genre.
- Word clouds and top-10 unigram counts for each genre are supplied in the appendix.
- The causal relation decision tree was removed from the appendix since we have uploaded and linked the guidelines.
- Numbers in the results section are slightly different because some of the data was missing from the original analysis. The trends and conclusions are not affected, only minor differences in the numbers.
- Some rewriting has been done throughout to shorten the sentence and improve readability.

Reviewer 2 Report

The paper describes a corpus of news, children stories and reddit posts, annotated on multiple layers: coreference, causal, temporal. The inter-annotator agreement has been performed and the values are usually reasonable (one example being alpha for temporal relations, a bit low).

The corpus is a very interesting resource for researchers who pursue multiple layers of text semantics. It opens up an opportunity to test multi-layered automated texts processing. The corpus is publicly available on github which is great news.

I have no critical remarks regarding the paper: it is well written, analyses are well designed, and inter-annotator agreement computed.

One drawback is modest size: the amount of annotated texts is on the level of hundreds, which can be borderline low to train large models. Best results are typically achieved using the level of thousands or tens of thousands of documents. Nevertheless, the corpus should be fine for testing.

Author Response

Changes:
- Added a paragraph to discussion to address adding a cross-linguistic dimension to the corpus.
- Added bar graph Figure 4 to section 5.3.2.
- Added explanation that Reddit comments are not included with the posts (end of section 4.1).
- Information is added about multiple DCTs introduced during updates or edits (section 5.2.2). 
- An extra column is added in Table 9 to distinguish event times and DCT times.
- Table 4 is added. This shows the average number of tokens and number of sentences per document for each genre.
- Word clouds and top-10 unigram counts for each genre are supplied in the appendix.
- The causal relation decision tree was removed from the appendix since we have uploaded and linked the guidelines.
- Numbers in the results section are different because some of the data was missing from the original analysis. The trends and conclusions are not affected, only the numbers.
- Some rewriting has been done throughout to shorten the sentence and improve readability.

Reviewer 3 Report

Exploring a Multi-Layered Cross-Genre Corpus of Document-Level Semantic Relations

This paper presents a multi-layer cross-genre corpus containing causal relations, temporal to leverage broader textual characteristics and semantic information.

I would equations if reddit can be used in combination with other sources. Are the data cured in terms of some target-specific topics? 

An word cloud or Tsne of these words as graphical or barplot distribution would be nice to see

How the high frequency of entities and characters unbalances the corpora. A plot of distribution regarding the topic themes would be beneficial to highlight some specifics of the reddit topics. 

It's known that first-name pronouns are used very often and generic ones. Hpw this affects usage by NER models for example?

The time annotation is the date of the first post or the replies of each subject topic? Perhaps establishing how this is incorporated would be beneficial.

I suggest the inclusion of a figure to represent one hirarquichal examples with the new and word classifications

Worldcloud or barplot word distribution in figures are needed to better assess the corpora, as well as the mains specs

The English is ok, More reduced to the point  sentences are welcome.

Sentences need to be reduced

Author Response

Response:

"I would equations if reddit can be used in combination with other sources. Are the data cured in terms of some target-specific topics? "
- Re: "Are the data cured in terms of some specific topics?". Yes, posts are about college from the college subreddits listed in the link provided.
- Re: "I would equations if reddit can be used in combination with other sources". We do not understand what this means, but our corpus does indeed use Reddit in combination with other sources. It is central to the paper.

"An word cloud or Tsne of these words as graphical or barplot distribution would be nice to see"
- We are not sure what words "these words" refer to. We have added word clouds and top-10 unigram counts for each genre to the appendix and we have added a bar graph to show the distribution of cause/enable/prevent across categories.

How the high frequency of entities and characters unbalances the corpora. A plot of distribution regarding the topic themes would be beneficial to highlight some specifics of the reddit topics.
- We are unsure what is meant by "How the high frequency of entities and characters unbalances the corpora."
- All of the reddit posts used in the corpus are about college.

"It's known that first-name pronouns are used very often and generic ones. How this affects usage by NER models for example?"
- We are confused about this comment. When the reviewer says "first-name pronouns", we assume the reviewer means first-person pronouns. What does it mean to say that they are used very often? The analysis shows that they are used with different frequencies in different genres.

"The time annotation is the date of the first post or the replies of each subject topic? Perhaps establishing how this is incorporated would be beneficial."
- Yes, this was not made clear enough in the original paper. The reddit posts do not include the replies/comments. We have made this clear. 
- Some Reddit posts have multiple DCTs---edits and updates are anchored to the time of the edit/update not the original creation time of the document. This has also been made clearer.

"I suggest the inclusion of a figure to represent one hierarchical examples with the new and word classifications"
- The reviewer does not explain what hierarchical examples this is in reference to or even what they mean by "hierarchical" here, nor do we understand what "the new and word classifications" means. Without clarification or elaboration, we are unable to respond to this comment.

"Worldcloud or barplot word distribution in figures are needed to better assess the corpora, as well as the mains specs"
- We thank the reviewer for this suggestion. We have added word clouds and top-10 unigram counts to the appendix. We make reference to these in support of our claims where appropriate.
- We are not sure what is meant by "main specs" here. We have added the average token and sentence length for documents in each genre, as well as total tokens and sentences per genre and tokens per sentence.

The English is ok, More reduced to the point sentences are welcome. Sentences need to be reduced
- We have tried to reduce the sentence length and improve readability throughout.

Changes:

- Added a paragraph to discussion to address adding a cross-linguistic dimension to the corpus.
- Added bar graph Figure 4 to section 5.3.2.
- Added explanation that Reddit comments are not included with the posts (end of section 4.1).
- Information is added about multiple DCTs introduced during updates or edits (section 5.2.2). 
- An extra column is added in Table 9 to distinguish event times and DCT times.
- Table 4 is added. This shows the average number of tokens and number of sentences per document for each genre.
- Word clouds and top-10 unigram counts for each genre are supplied in the appendix.
- The causal relation decision tree was removed from the appendix since we have uploaded and linked the guidelines.
- Numbers in the results section are different because some of the data was missing from the original analysis. The trends and conclusions are not affected, only the numbers.
- Some rewriting has been done throughout to shorten the sentence and improve readability.

Round 2

Reviewer 3 Report

I suggest the authors to graphically establish the relation the specify  genre-specific characteristics on each of de defined layers, see what is common, what is unique, for example, a 10-word distribution of each layer, and how they intersect or are distinct.

Also, e more focused to the point introduction to better understand in clear way the main problem to be addressed or methodology. It is not clear or in other words, has to much text cluttering the main objective.

Minor

Author Response

"I suggest the authors to graphically establish the relation the specify  genre-specific characteristics on each of de defined layers, see what is common, what is unique, for example, a 10-word distribution of each layer, and how they intersect or are distinct."

- We thank the reviewer for this suggestion. We have added a visualization in Figure 5. The figure represents the most prominent semantic characteristics of each genre type across the three semantic layers. We want to thank the reviewer for pushing us to include this figure since we feel it helps the reader better appreciate the differences being described.

"Also, e more focused to the point introduction to better understand in clear way the main problem to be addressed or methodology. It is not clear or in other words, has to much text cluttering the main objective."

- We have tried to rewrite the introduction to address this comment. We have reorganized the paragraphs, and made the sentences shorter and less wordy. We have also shifted the language from presenting the work as being descriptive to presenting the work as analytical. While the work has a significant descriptive component, it is only as a result of careful, well constructed quantitative analysis.